# Structural Damage Detection Based on the Correlation of Variational Autoencoder Neural Networks Using Limited Sensors

**DOI:** 10.3390/s24082616

**Published:** 2024-04-19

**Authors:** Jun Lin, Hongwei Ma

**Affiliations:** 1School of Environment and Civil Engineering, Dongguan University of Technology, Dongguan 523808, China; ivorlin@aliyun.com; 2Guangdong Provincial Key Laboratory of Intelligent Disaster Prevention and Emergency Technologies for Urban Lifeline Engineering, Dongguan 523808, China; 3School of Aerospace, Xi’an Jiaotong University, Xi’an 710049, China

**Keywords:** structural damage detection, variational autoencoder neural networks, limited sensors, correlation, structural health monitoring

## Abstract

Identifying the structural state without baseline data is an important engineering problem in the field of structural health monitoring, which is crucial for assessing the safety condition of structures. In the context of limited accelerometers available, this paper proposes a correlation-based damage identification method using Variational Autoencoder neural networks. The approach involves initially constructing a Variational Autoencoder network model for bridge damage detection, optimizing parameters such as loss functions and learning rates for the model, and ultimately utilizing response data from limited sensors for model training analysis to determine the structural state. The contribution of this paper lies in the ability to identify structural damage without baseline data using response data from a small number of sensors, reducing sensor costs and enhancing practical applications in engineering. The effectiveness of the proposed method is demonstrated through numerical simulations and experimental structures. The results show that the method can identify the location of damage under different damage conditions, exhibits strong robustness in detecting multiple damages, and further enhances the accuracy of identifying bridge structures.

## 1. Introduction

Scholars across diverse disciplines have extensively researched and experimentally validated damage identification methods for Structural Health Monitoring (SHM) [1]. Bridge SHM is fundamentally about identifying, detecting, and diagnosing the true structural condition, which is crucial for research in structural damage detection and identification [2]. Scholars commonly utilize Frequency Response Function (FRF) [3] and structural vibration modes [4] for structural state identification. Nevertheless, these methods necessitate relatively complete structural modes. Consequently, scholars are further researching feature extraction methods [5] with or without reliance on models.

The methods for identifying structural damage can be broadly categorized into model-based methods and model-free methods. Model-based methods depend on model correction, involving the use of finite element models to align predicted responses with measured responses, thereby identifying structural damage. Roy et al. examined alterations in mode shape for damage detection [6]. Cowley and Adams pioneered research in 1979 on detecting damage through variations in a structure’s natural frequency [7]. Another type of model-based method involves the early development of shallow artificial intelligence methods, such as Artificial Neural Networks (ANN) [8] and Back Propagation (BP) algorithms [9], to identify unreliable factors in the system based on existing observational data, historical data, and current observations. Shu et al. employed a simplified railway bridge model alongside artificial neural networks for structural damage detection [10]. Zhang et al. analyzed the structural damage diagnosis of the bridge behind the abutment with the BP neural network [11]. Model-based methods rely on numerical models, assuming the structure is in a good initial state, necessitating highly accurate modifications to the model. Hence, model-based damage identification methods must establish a relatively accurate model [12]. Xiao et al. Using a multi-scale model, the structural state under different temperatures and stiffness levels was analyzed [13,14,15]. Komarizadehasl S. et al. using new damage identification for damaged reinforced concrete structures [16]. However, in actual bridge structures, the complexity of various bridges and monitoring systems makes it challenging to modify real structures effectively and establish detailed finite element models. Therefore, model-free methods based on vibration measurement signal data, without relying on finite element models, are more suitable for practical engineering applications.

A model-free approach that does not depend on finite element model analysis but directly computes the structure’s response, employing the classic “data-driven” algorithm Multilinear Principal Component Analysis (MPCA) [17], by extracting the structure’s feature information to determine its state. In addition to the above methods, deep learning neural networks are also data-driven and exhibit model-free method characteristics. Damage can be identified by constructing specific networks to extract structural features [18]. Effectively employing deep learning methods for structural damage identification remains a significant research direction. Presently, research in this field can be roughly divided into two categories: one is based on computer image recognition technology to identify and classify structural images for damage identification [19]; the other method involves using deep neural networks to extract damage-sensitive features from the structure’s dynamic response of damage labels and predict the location of structural damage [20]. Xu et al. classified and located the damage of various types of reinforced concrete bridge piers subjected to earthquakes by constructing a deep learning Convolutional Neural Network (CNN) model [21]. Deng et al. utilized computer vision technology and consumer-grade cameras to process crack images or civil structures [22]. However, these two methods still have specific computational efficiency requirements and limitations in computational efficiency and actual structures. Lin et al. utilized Convolutional Neural Networks (CNN) to automatically extract features from original time-domain signals of structures and identify damage locations, presenting a novel avenue for research in structural modal identification [23]. Duan et al. used spatial-spectral information with convolutional neural networks to detect the damage of tied-arch bridges [24]. Osama et al. developed a one-dimensional Convolutional Neural Network (CNN) architecture integrating feature detectors and classifiers to successfully detect damage in a model of a sports stadium grandstand [25]. The variational autoencoder neural network employed in this study is an unsupervised learning technique capable of identifying latent data features in unlabeled datasets [26]. Common unsupervised learning methods include autoencoder neural networks and generative adversarial neural networks [27]. 

To enhance computational efficiency and accuracy in damage detection, this paper’s method relies on correlation and Variational AutoEncoder (VAE) to leverage the benefits of unsupervised learning for automatic feature extraction from signals, enabling swift and efficient processing of extensive bridge monitoring data. VAE, an unsupervised learning method based on variational autoencoders, comprises an encoder structure with input, hidden, and output layers [28], similar to the traditional autoencoder. Researchers have categorized different network architectures based on model requirements while maintaining the overall framework. Experimental analysis of a specific model demonstrates that the recognition method based on VAE can effectively accomplish the identification task, with the network model aiming to replicate input samples as closely as possible to capture implicit features representing the input sample characteristics. The model used by Louizos et al. is based on a variational autoencoder architecture with priors encouraging independence between sensitive and latent change factors [29].

This article proposes a damage detection method for a limited number of sensors based on correlation using a Variational Autoencoder (VAE) [30] due to the challenges in implementing traditional signal analysis methods that require non-destructive data for comparison as the basis in practical engineering. The method leverages unsupervised learning by automatically extracting features from signals, enabling the swift and effective processing and analysis of large amounts of data obtained from bridge monitoring without the availability of baseline data for comparison.

The subsequent sections of this paper are structured as follows: Section 2 provides a brief overview of autoencoders and variational autoencoder neural networks, along with the development of data augmentation and damage factors. Section 3 delineates the network architecture of this approach and confirms the viability of the VAE damage identification method through finite element models. Section 4 extends this validation using experimental models. Lastly, Section 5 systematically summarizes the content of this paper.

## 2. Methods

This section mainly elaborates on the relevant theories and the construction of damage factors in this research method, including the architecture and introduction of autoencoder neural networks and variational autoencoder neural networks.

### 2.1. Artificial Neural Network and Autoencoder

The classical artificial neural network (ANN), a mathematical model architecture constructed by simulating biological neural networks and other network systems, possesses analytical structure and recognition system functions. It generally consists of a certain number of interconnected neural nodes [31]. A simple artificial neural network model system is a nonlinear dynamic network structure composed of a large number of highly parallel and interconnected simple nonlinear processing nodes, with each neural node in the network referred to as an activation function [31]. Mature artificial neural networks can adaptively adjust the learning structure’s feature information, exhibiting good nonlinear recognition capabilities and robustness, with the most significant feature being adaptability, i.e., the neural network’s self-regulation ability, including learning, training, self-organization, and generalization capabilities [32]. The neural network can utilize the system’s built-in algorithms to identify the transformed signal weight values, forming the memory of the artificial intelligence neural network. Artificial neural networks logically express certain signal features by recognizing the feature information of the structure, but artificial neural network systems generally involve an approximate solution process.

Unlike ANN, autoencoder neural networks place more emphasis on feature extraction of information. Researchers widely use autoencoder neural networks as a type of neural network model based on unsupervised learning. The core of the autoencoder in autoencoder neural networks is to have the output values equal to the input values. The network structure of autoencoder neural networks is similar to that of deep learning networks, consisting of input, hidden, and output layers. However, autoencoder neural networks utilize constrained hidden layer feature values for systematic research and analysis, ensuring that the number of hidden layer features matches the number of input values, resulting in the same number of output layer neurons as input layer neurons [32]. The fundamental idea of autoencoder neural networks is to use a large amount of known sample information to predict the network’s output results. They excel at handling high-dimensional nonlinear systems, making them widely used in areas such as system pattern recognition and data signal information mining [32]. In principle, autoencoder neural networks project the original data onto several unrelated linear orthogonal axes to obtain the data’s feature representation, allowing for the reproduction of input signals as much as possible.

As depicted in Figure 1, an autoencoder is divided into an encoder and a decoder. The labeled dataset X is input into the encoder, and through training to adjust network structure parameters, such as restricting the number of nodes in the hidden layer, the output X’ is obtained. Therefore, the autoencoder essentially fits an identity function, with the number of nodes in the hidden layer being fewer than the number of nodes in the input layer, enabling the hidden layer’s feature information to represent the input data effectively. During this process, algorithms like Backpropagation (BP) and Levenberg-Marquardt with Conjugate Gradient (LM-CG) can be employed to complete the training of the hidden layer. To enhance the mapping capability of neural networks, it is often necessary to establish deep autoencoder neural networks with more than five network layers. However, this poses the risk of network overfitting, where the network may simply replicate the input as output, and the encoding layer may not effectively represent the data. Therefore, certain constraints must be set in the encoding layer to address the problem-solving needs.

### 2.2. Variational Auto-Encoder

Variational Autoencoder (VAE) addresses the limitations of autoencoder neural networks, which cannot generate data on their own and struggle to determine the accurate distribution of feature information in the hidden layers. By introducing a hidden variable Z in the hidden layer and controlling its distribution, VAE aims to make the output controllable [33]. The study explores various statistical feature extraction algorithms and ultimately focuses on the Deep Bayesian Network for analysis. This network can represent relationships between variables using neural networks and effectively analyze complex structured data to identify feature information accurately.

VAE is a probabilistic model based on variational inference, aiming to establish a generative model rather than just an image network. By extracting feature information through an approximate model function and reducing errors, VAE enhances computational efficiency. The text emphasizes the importance of imposing constraints on the network to ensure convergence during training and prevent potential variables from affecting final prediction results. By adding specific conditional restrictions, VAE overcomes the drawbacks of autoencoder neural networks and reflects the relationship between hidden variable Z and visible variable X. Overall, VAE generates effective and reasonable feature information within the network, with the visible variable X being generated by the hidden variable Z. As shown in Figure 2, with z following a Gaussian distribution N(0,1). Sampling z from p(z), data is auto-generated through $p_{\theta }(x|z)$. Thus, the observable variable x is generated by the latent variable z, and z→x represents the generative model $p_{\theta }(x|z)$, which, from the perspective of an autoencoder, acts as the decoder. $p_{\theta }(x|z)$ can be implemented through neural networks. x→z is the recognition model $q_{\varphi }(z|x)$, similar to an encoder in an autoencoder. The overall structure of a VAE, as shown in Figure 3, differs from a standard autoencoder in that VAE imposes additional constraints on the hidden layer, making it controllable.

After obtaining p and q, in order to achieve a good result, q needs to be as close to p as possible. The key is to measure the gap between *q* and *p*. Variational Autoencoder (VAE) uses KL divergence to measure the difference between *q* and *p*. A smaller KL value indicates a closer distance between them. In VAE, estimation of the parameters of the generative model is required, thus assuming an unknown distribution that satisfy the following relationship:(1)$$D_{KL}(p(x)\left\|q(x)\right.)=-\int p(x)\log q(x)dx-(-\int p(x)\log p(x)dx)=\int p(x)\log\left\{\frac{p(x)}{q(x)}\right\}dx$$

The above Formula (1) is called the relative entropy, Kullback-Leibler divergence, or KL divergence between p(x) and q(x). Since the above formulas are not symmetric in structure, $D_{KL}(p\left\|q)\neq D_{KL}(q\left\|p)\right.\right.$.

By deriving the formula from the previous section, it can be seen that the Variational Autoencoder (VAE) needs to reduce the gap between *p* and *q*. In practical applications, for the latent variable z in the hidden layer, $p_{\theta }(x|z)$ will be 0, $p_{\theta }(x)$ represents the distribution that needs to be satisfied when generating the data set *x*, so the calculation of $p_{\theta }(x|z)$ with $p_{\theta }(x)$ does not affect it. The core idea of VAE is to sample z and calculate $p_{\theta }(x)$ from the sampled z, $E_{q_{\phi }(z|x)}p_{\theta }(x|z)$ should be as closely related to $p_{\theta }(x)$ as possible. The Kullback-Leibler divergence between $p_{\theta }(z|x)$ and $q_{\phi }(z|x)$ is:(2)$$D_{KL}\left[q_{\phi }(z|x)\left\|p_{\theta }(z|x)\right.\right]=E_{q_{\phi }(z|x)}\left[\log q_{\phi }(z|x)-\log p_{\theta }(z|x)\right]$$

Applying Bayes’ rule to $p_{\theta }(z|x)$, substituting $p_{\theta }(x|z)$ and $p_{\theta }(x)$ into Formula (2) allows us to transform the KL divergence into:(3)$$\log p_{\theta }\left(x\right)-D_{KL}\left[q_{\phi }\left(z|x\right)\left\|p_{\theta }\left(z|x\right)\right.\right]=E_{q_{\phi }\left(z|x\right)}\left[\log p_{\theta }\left(x|z\right)\right]-D_{KL}\left[q_{\phi }\left(z|x\right)\left\|p_{\theta }\left(z\right)\right.\right]$$

Formula (3) is the foundation of VAE. In order to make *q* as close to *p* as possible, the KL divergence should be minimized. Therefore, the left side of the equation should be maximized. When $q_{\phi }(z|x)$ is given, a stochastic gradient descent algorithm is used to optimize the right side. Therefore, instead of relying on z, X is predicted through training $q_{\phi }(z|x)$. VAE can compress high-dimensional data into low-dimensional z, and then the generative network will produce a distribution that is as similar as possible to the original data.

### 2.3. Data Preprocessing, Augmentation, and Moving Window Settings

The variational autoencoder neural network method used in this article requires a certain dataset for training. Since only a small amount of sensor data is utilized, data augmentation is needed for the data obtained in this article. Additionally, this section will provide some explanation of the construction of damage factors.

#### 2.3.1. Data Preprocessing

Due to the unit restrictions of the obtained data itself, all the simulated data in this paper are acceleration data. Therefore, the obtained data needs to be normalized. Normalization transforms the response signals obtained into dimensionless data to avoid the inability to compare and weigh the feature information of different data or magnitudes due to different unit restrictions during network model analysis. Data preprocessing allows dimensionless or data with different units to be better compared without affecting the intrinsic feature information of the data. Normalization is a crucial step in the data preprocessing stage using deep learning algorithms. Since the data obtained itself has unit restrictions, all the data simulated in this paper are acceleration data. Therefore, it is necessary to normalize the obtained data. Normalization transforms the acquired response signals into dimensionless data to avoid the inability to compare and weigh different data or feature information of different units during network model analysis. Data preprocessing allows dimensionless or data with different units to be better compared without affecting the intrinsic feature information of the data. The normalization formula used in this method is the Atman standardization formula as follows:(4)$$\bar{x}=\frac{x-\mu }{\sigma }$$
where x is the sample set, μ is the mean of sample x, and σ is the standard deviation of sample x. The data samples are compressed into the range of 0 to 1 to simplify the response data.

#### 2.3.2. Data Augmentation

Based on the method of variational autoencoders, after data preprocessing, it is essentially based on neural network artificial intelligence. Therefore, more data is needed for training and learning neural networks to achieve better generalization effects. Therefore, it is necessary to increase the diversity of the dataset. However, obtaining more data in practical engineering means that structural monitoring systems require more resources, and it is not possible to obtain various types of data in practical engineering monitoring. Moreover, this method only utilizes a small amount of sensor data for research and analysis. Therefore, existing small amounts of sensor response signals need to be augmented to improve the diversity of data obtained by using data augmentation methods on the existing data. By training the augmented dataset, the variational autoencoder network model can learn the structural feature information in a more diverse way, thereby improving the recognition and generalization capabilities of the network model. Specific data augmentation methods should be set according to the goals to be achieved, defining the characteristics of the dataset to make the dataset more diversified.

In the variational autoencoder neural network applied in this paper, the response signals obtained by a small number of sensors need to be augmented to alleviate the lack of data in the variational autoencoder neural network learning algorithm. The damage conditions obtained from the finite element model of simply supported beams under moving loads and simulation experiments in this paper are limited, leading to an insufficient training dataset to support the learning and training of the network. 

For structural health monitoring problems, the structural responses measured by sensors can be viewed as continuous long-time series. The data augmentation method used in this paper randomly selects *m* test values from the entire initial dataset as initial time series data of *m*, then randomly truncates time series segments to define *m* short test segments of a certain length and adds them as samples different from the original structural sample data to the training data batch of the structural network model. Specifically, in each iteration of the neural network, a certain number of data samples are selected to calculate the current gradient of the neural network, and this sample set is called a data batch. After augmenting the information from a small number of sensors, the data are divided into training, validation, and testing sets. The augmented data still retains the structural feature information, so 60% of the augmented data is used as the training set, and the remaining 40% is equally divided into the validation and testing sets. After the training sample set is exhausted, a new input vector and output vector are reconstructed for each previously selected sample. This process results in the final network parameters and corresponding performance evaluation metrics. In the method studied in this paper, the data testing set of the simulation experimental model is different from numerical simulation. The acceleration response data obtained from the simulation experiments are abundant, and each damage condition measurement is repeated five times. Therefore, any three sets of the acceleration response information obtained from the simulation experiments are used as the training set for network training, while the other two sets are used as the validation and testing sets. Data augmentation needs to be performed after standardizing the acceleration response data. The data augmentation process in this study is shown in Figure 4.

#### 2.3.3. Setting of Moving Window

By preprocessing and expanding the data, the numerical simulation and experimental dataset size meet the neural network training needs. However, it is still necessary to classify the dataset and set the window length so that the variational autoencoder neural network can learn and train correctly. Therefore, a method of extracting collected data reflecting information through a moving window is proposed. Assuming the length of the window can be determined based on the characteristics of Shannon’s theorem and the structural frequency, the window length *l* is denoted as:(5)$$l\geq 2\frac{f_{s}}{f_{1}}$$

The sampling frequency is denoted as $f_{s}$, and $f_{1}$ represents the fundamental frequency of the response. The Fast Fourier Transform (FFT) [29] is applied to convert the training sample into the frequency domain signal to determine the fundamental frequency.

#### 2.3.4. Setting and Detection Steps of Damage Index

Due to the redundancy of the initial acceleration signal, the sudden change value of the damage cannot be directly observed. Therefore, a damage index is established to represent the extracted damage information. If the bridge structure is damaged, the vibration characteristics of the bridge change and the corresponding acceleration signal input to the Variational Autoencoder (VAE) neural network will also alter the structural feature vectors. The feature vectors obtained before and after the structural damage are different [34]. Hence, the damage-sensitive factor can be set as the correlation between the feature vectors of adjacent windows to determine the state of the bridge structure:(6)$${\mathrm{CI}}_{i}=\rho (T_{i},T_{i+1})$$
where ρ is the correlation coefficient, the larger the value of the correlation coefficient, the more stable the structural position state. If the value is smaller, it indicates a poorer correlation and the structural position has experienced abnormal damage.

## 3. VAE-Related Correlation Analysis Numerical Simulation

According to the author’s research findings, when a moving load passes through a damaged location on a bridge, it leads to a deterioration in correlation at that point [34]. The variational autoencoder neural network can extract key feature information from the acceleration response signals of a few sensors, enabling the determination of the damage location in a beam-type structure through correlation analysis of these key features. In this chapter, numerical simulations of a simply supported beam model under a moving load are conducted using finite element methods. By analyzing and training the variational autoencoder neural network with the acceleration response signals from a few sensors placed on the simply supported beam bridge, abrupt changes in correlation in the signals are identified to determine the location of the damage. Acceleration data from simulations are preprocessed and augmented through windowing to obtain training samples. The acceleration responses from numerical simulations are trained using the optimization algorithm of the variational autoencoder neural network, primarily utilizing the Keras module for training. Subsequently, the feature information from the hidden layers of the variational autoencoder is extracted and analyzed for correlation to determine the location of the damage in the beam-type structure.

### 3.1. Finite Element Model and Parameter Settings

This section mainly focuses on the finite element simulation beam’s working conditions and parameter settings.

#### Finite Element Model Parameters

The variational autoencoder neural network method studied in this chapter analyzes the damage conditions of numerical simulations. The numerical simulation of the simply supported beam bridge uniformly sets up seven measurement points to obtain acceleration data responses, as shown in Figure 5. The finite element parameters are as shown in Table 1. Damage ratio γ (the ratio between damage depth and h) for 10%, 20%, 30%, 40%, and 50%. The speeds of the trolley are 0.25 m/s and 0.5 m/s, the mass of the moving trolley is 200 kg, and the sampling frequency of the numerical simulation is 200 Hz. Each working condition uses the dataset composed of the acceleration data responses measured by the predetermined four acceleration sensors. This method analyzes single damage conditions (SDC), multiple damage conditions (MDC), and situations affected by noise, as shown in Table 2 and Table 3 above.

### 3.2. Finite Element VAE Network Configuration

This section mainly focuses on constructing and analyzing network architecture under numerical simulation conditions. The overall architecture of the VAE network is divided into three parts: training, testing, and validation. After data augmentation of the acceleration signals from a few sensors using the calculated specific length window, the data within the window is of consistent length. This data is then randomly combined and arranged to generate a training dataset for use in the variational autoencoder network model. Subsequently, the VAE is trained on this dataset, and model parameters are adjusted so that the network model can learn effective feature information from the data. After training the well-architected network model, the acceleration response data measured by the aforementioned few sensors is cut and arranged, with the remaining 40% serving as the test and validation sets. However, the time series of the validation and test sets must not be altered because the feature information contained in the acceleration response signals includes correlation information related to structural damage. A hidden encoder in the variational autoencoder neural network can maintain the same time order as the hidden encoding to understand the original data’s basic characteristics. After learning and training through the variational autoencoder neural network, the feature information of the hidden layer is extracted. The damage factor based on the correlation analysis method of a few sensors using the variational autoencoder neural network is determined by analyzing the correlation of feature information. The damage factor reflects the correlation of structural feature information, and the state of the structure is judged based on the compatibility of the correlation. This network model structure does not require undamaged baseline data of the beam structure response; it only uses the acceleration response data of the simply supported beam structure under damaged conditions measured by a few sensors to train the well-architected VAE model without needing to obtain inherent parameters of the structure from the complete structure.

#### 3.2.1. Setting of Moving Window Length

According to Section 2.3.3 above, performing windowed calculations and processing the data is necessary. Therefore, the fundamental frequency of the calculation model needs to be determined. The structure’s frequency can be obtained using the Fourier transform with simulated data in this section.

Figure 6 shows the Fourier spectrum of the acceleration response of the undamaged beam at sensor 1. According to the figure, the fundamental frequency of the structure is 1.123 Hz. Shannon’s theorem and the window function formula define the window length l as 357. Since the data length of the input layer of the variational autoencoder neural network is generally a power of 2, and to ensure the window length is greater than 357 and fits the network structure and data length appropriately, this paper sets the window length to 1024, meaning the dimension of the input layer data for the training set is (1024,4). Moreover, by controlling the calculation step size to reduce training time, a step size that is too small would lead to excessively long training and computation times; hence, this paper sets the step size to 32.

#### 3.2.2. Network Model Parameter Settings

The network layers of the variational autoencoder neural network are shown in Table 4, where each layer is followed by a normalization process, with Leaky ReLU serving as the network’s activation function.

During the training process, it is necessary to use some hyperparameter techniques to improve the training process, such as reducing the learning rate, saving the model when the loss value reaches a limit, and then directly searching for the mode with the lowest loss rate. For non-minimum phase problems, early stopping parameter techniques are a good solution. A hyperparameter technique can reduce training time by immediately stopping the neural network model training when the loss function decreases and stabilizes.

The VAE network determines the input data structure based on the window length and sets the network parameters according to the structure’s fundamental frequency. It uses data from a few sensors to determine the basic frequency of the response signal and window length. The network parameters are fine-tuned based on research by scholars on this network, and appropriate model training is conducted. This method selects four sensor measurement points for calculation; hence, the input dimension is (1024,4). The overall architecture of the variational autoencoder neural network is shown in Table 4 above. The architecture of the network includes encoders and decoders with different convolutional layers. After convolution, the convolution results of the input data still need further feature extraction. Therefore, this network architecture uses the max pooling layer to extract features from the information further after the convolutional layer. This dataset is the convolutional pooling extracted dataset from the input signal, which still needs to go through the feature extraction process of the hidden layer. Therefore, the dimension of the hidden layer is set to 428. In the variational autoencoder neural network, the deconvolution layer (Conv1D) of the decoder follows immediately after the fully connected layer, but a pool is needed as a supplement during decoding. In order to make the input data and output data structures the same in the decoder structure, this method uses deconvolution layers and unpooling layers to reconstruct the data. The activation function uses Leaky ReLU combined with batch normalization to reduce the Internal Covariate Shift (ICS) effect in the variational autoencoder neural network. The ICS effect is where the distribution of input and output signals changes due to network parameter updates. Therefore, by combining the activation function with the reduction of the network’s ICS and the control of gradient dispersion’s KL divergence effect, the training loss value is effectively reduced without increasing the direction of distribution change due to parameter settings.

During the training process, the adaptive optimization algorithm RMSprop is used, introducing the gradient direction of historical information parameters to avoid non-convergence or serrated descent of the gradient, making the parameter optimization process more stable and quickly converging to the global optimum. Finally, the RMSProp function is applied in the variational autoencoder neural network, allowing all parameters in the network to be optimized quickly and converge to the global optimal state. The RMSProp function can choose different learning rates for different parameters. It only needs to set the learning rate before network training, and then the algorithm automatically adjusts as follows during the training process:(7)$$v_{t}=pv_{t-1}+(1-p)g_{t}^{2}$$
(8)$$\Delta w_{t}=-\frac{\eta }{\sqrt{v_{t}+\varepsilon }}*g_{t}$$
(9)$$w_{t+1}=w_{t}+\Delta w_{t}$$
(10)$$\eta_{i+1}=\eta_{i}*\alpha$$

The Formulas (7)–(9) mentioned are for the RMSprop algorithm, where η is the initial learning rate, $v_{t}$ represents the exponential moving average of the gradient direction; $g_{t}$ represents the time gradient, *p* is the decay factor (usually set to 0.9), used to control the decay rate of historical information, and ε is a small constant to avoid division by zero.

The Formula (10) mentioned is the learning rate formula. In the variational autoencoder neural network method, the value of α is set to 0.1, and the learning rate is set to 0.001. The outcome of the network training is judged based on the observed trend of the loss function’s decline.

### 3.3. Results of Numerical Simulation

By optimizing the algorithm and modifying the loss function, the simply supported beam model can be analyzed. This study introduces a threshold setting method to determine whether the fluctuations and peaks at a certain point are abnormal, preventing the network model from misjudging. The threshold is set as described in Formula (11), with the value for both numerical simulation and experiment set to 98%. The well-constructed network is used to test and analyze the damage conditions and to identify the damage factor. Herein, the unilateral confidence limit (UCL) [35,36,37] is introduced, and the threshold can be given as:(11)$$T^{\alpha }=\mu +Z_{\alpha }\sigma$$
where $T^{\alpha }$ is the confidence limit, μ and σ are the mean and standard deviation of the damage factors calculated after this method, respectively, and $Z_{\alpha }$ is the value of the standard normal distribution with a mean of zero. The variance represents the value of the beam following a standard normal distribution, ensuring that the cumulative probability of threshold selection is $\left(1-\alpha \right)\times 100$%. In this study, a threshold of 2% is chosen, indicating that 98% of the VAE damage factors fall below this confidence limit. This threshold serves as a critical value for the damage indicator. When the damage indicator surpasses the threshold, it signifies that damage has occurred at that specific location. By considering the statistical properties of the calculated damage factor, the damage location can be identified by comparing the damage factor to the threshold value. Each damage scenario yields a distinct threshold, and the threshold lines displayed in this paper are calculated based on the minimum damage state.

Since the correlation coefficient is a value between [0, 1], and a larger value indicates good correlation at a certain point, if identifying the damage location, it is preferable for the damage indicator derived to be as small as possible. If the indicator is below the threshold, it indicates an abnormal condition at that location. The indicator values calculated by numerical simulation and experiment are all smoothed.

Although this method does not require baseline data, for the purpose of comparison, an analysis is first conducted using a set of undamaged data, as shown in Figure 7. To accurately represent the location of the damage, the x-axis adopts the method of relative position, and the y-axis is the correlation damage indicator. It can be seen that under undamaged conditions, the correlation damage indicator tends to be stable and close to 1, indicating that the network has a good correlation with the input data and the structural state is stable without abnormalities.

#### 3.3.1. Single Damage Condition

Based on the above numerical simulation conditions, the trained network is used to analyze and calculate the single damage condition. As shown in Figure 8 and Figure 9, it can be seen that under conditions of either 0.2 m/s or 0.5 m/s, compared to the undamaged situation, a 10% damage at 0.4 L shows a problem of decreased correlation, exceeding the threshold indicator, and accurately locating the position of the damage. For the overall 10–50% damage condition graph, the greater the damage, the smaller the correlation coefficient determined by the network for the damage location, indicating a worse correlation. This means the degree of damage is reflected in the network’s indicators as lower correlation coefficients, the greater the degree, and the network identifies the place of damage as the location with the worst correlation. At the same time, the location of damage can be identified under different speeds, and when the speed increases and the data volume decreases, the variational autoencoder neural network can still analyze the structural condition effectively.

In addition to the 0.4 L damage location, this method also conducted computational analysis for different damage locations. Using the VAE method to continue analyzing the damage condition at the 0.7 L position, it can be seen in Figure 10 below that the VAE network’s correlation analysis indicators can identify and judge the structural state and damage location under different damage conditions. Although there are some values at other locations that do not approach 1, the overall trend shown by the curve is not significantly different, and the indicator numbers at 0.7 L all show a sudden change and result in a worse correlation; the VAE network can accurately identify the location of damage in different positions.

Since the numerical simulation results will not have noise and vehicle-bridge coupled vibration due to external factors such as road surface roughness affecting the signal change, it is possible to determine the damage location as the place of sudden change under the judgment of damage indicators and thresholds, proving the method to be effective. To assess the robustness of the identification method against noise, this study also analyzed numerical simulation acceleration data under different signal-to-noise ratios (SNRs). Gaussian white noise is superimposed on the original signal, and the signal-to-noise ratio (SNR) used to quantify the noise level can be defined as [38,39]:(12)$$SNR=10\mathrm{lg}\frac{P_{x}}{P_{N}}$$
where $P_{x}$ and $P_{N}$ are the powers of the original signal and noise.

Gaussian-distributed white noise was added to the acceleration response signals measured by a small number of sensors in the numerical simulations, with noise SNRs of 25 dB and 40 dB. The cart speed was still 0.2 m/s, and the damage location was at 0.4 L. Whether the SNR was at 40 dB or 25 dB, the numerical simulation of damage showed that the damage indicator was still possible to determine that the place with the worst correlation was still the damage location, and all were below the threshold. As shown in Figure 11, when the noise is amplified, this method can still effectively identify the state of the structure. At the same time, the network may be slightly disrupted, but it does not affect its ability to determine the location of structural damage. It can still be seen that the correlation analysis method of a small number of sensors based on the variational autoencoder neural network can identify and analyze the response signals under noisy conditions well.

#### 3.3.2. Multiple Damage Conditions

The previous section analyzed the recognition of single damage conditions by the variational autoencoder neural network, but in actual engineering projects, bridge structures generally do not have only one damaged or abnormal location. Therefore, multiple damage conditions also need to be validated through this method.

The simulation of two damage locations was conducted with one damage at 0.35 L of the beam length maintained at a 10% damage condition, while the damage severity at the 0.6 L location increased from 10% to 50%, with the cart speed remaining at 0.2 m/s in Table 3. The results are shown in Figure 12 below; damage at both 0.35 L and 0.6 L can be accurately identified, and both are below the threshold value, indicating poor correlation at the two damage locations. It can be clearly seen that when both damages are at 10%, the indicators of the two damage locations are basically the same and below the threshold, indicating that the network model identifies poor correlation at the damage locations, thus determining the structural state and that the structure has damage at both locations. As the damage severity at 0.6 L increases, its damage indicator decreases with increasing damage severity, and the magnitude of the decrease is significantly greater than that at 0.35 L, as shown in the figure. The summary graph of various damage conditions shows that the variational autoencoder neural network can still accurately identify the correlation of damage locations based on the damage indicators. The correlation is worst at the 0.35 L and 0.6 L damage locations, i.e., their damage indicators are lowest at the damage locations, and only the values at these two damage locations are below the threshold. This indicates that the model based on the variational autoencoder neural network can accurately identify these two damage locations, demonstrating that the model structure of the variational autoencoder neural network can well recognize both single and multiple damage conditions.

This method also analyzes three damage conditions to validate the network’s effectiveness under multiple damage conditions. Based on the original dual damage condition, as shown in Figure 13, the damage is added at 0.5 L from 10% to 50%, and the network is used for analysis and identification. It can be seen that the network can effectively identify the location of the structural damage, indicating that this method can identify multiple damage conditions well.

Through numerical simulation, it can be visually observed that the variational autoencoder neural network can effectively identify the state of the structure, including under different speeds, damage locations, and noise conditions. It can also determine the location of damage under multiple damage conditions, demonstrating good adaptability.

## 4. Experimental Verifications

The previous section mainly focused on validating the method using numerical simulation data, while this section further discusses and validates the experimental structural model data.

### 4.1. Experimental Setup

This section will use an experimental model for validation to verify the effectiveness of the method proposed in this paper. This section will focus on introducing the experimental conditions and apparatus.

The experimental beam uses an equal-section steel box beam as the test material for the simply supported beam bridge. The designed beam structure model uses hollow square steel as the main bridge. As shown in Figure 14 below, two symmetrical angle steel tracks are laid on both the model experimental beam bridge and the approach bridge, with a distance of 15 cm between the angle steel tracks. The designed mobile cart is installed on the angle steel tracks to simulate the coupled vibration generated by the moving vehicle load passing through the beam bridge and to acquire the signal characteristics of the bridge. The hollow square tube beam bridge is 6 m long and 0.2 m wide. In this experiment, the measurement data is based on a 6 m medium-span cross beam, and constant-speed bridges of 4 m and 2 m are set up at both ends of the simply supported beam bridge model, as shown in Figure 14b. Due to the traction force required by the moving vehicle, the vehicle accelerates slowly from a stationary state to a constant speed under the action of the traction force to ensure that the vehicle passes through the test bridge model at the mid-span at a constant speed, thereby obtaining effective acceleration signals.

In this model experiment, a self-designed bolted steel-hole cart is used for the moving load experiment. As shown in Figure 15a, the towed mobile cart has a mass of 10.5 kg. Iron blocks are used to increase the load and achieve high-weight conditions. The speed of the moving load cart is changed by adjusting the rotational speed of the motor through a gearbox. The traction rotational speed of the motor used in this beam bridge experiment is 100 r/min and 200 r/min, and the corresponding speed is calculated to be 0.25 m/s and 0.5 m/s. This model experiment utilizes an existing dynamic signal testing system for research. Data is collected and analyzed using a DH5922N dynamic signal acquisition device, as shown in Figure 15b below, with a sampling frequency of 500 Hz. Underneath the model bridge deck, 7 IEPE accelerometers are attached using magnetic stickers and are evenly distributed along the axial centerline of the beam to simultaneously obtain acceleration response signals.

The method proposed in this paper requires only a few sensors to measure and analyze acceleration. However, to ensure that sensors cover the key positions of the beam, seven acceleration sensors are distributed based on the length of the main bridge center, as shown in Figure 16 from points S1 to S7 from left to right. Using an electric angle grinder, a small crack is introduced on the section of the beam at a distance of 0.68 L from the left end of the main bridge, forming a damaged condition. Figure 17 shows images of the damage crack, including side and bottom fracture views. The conditions are as follows: single damage condition (SDC) and multiple damage conditions (MDC) are also considered, as shown in Table 5.

Using the acquired acceleration data, the effectiveness of the variational autoencoder neural network method for correlating a small number of sensors is verified. The simulation experiment obtained a large amount of acceleration response data, and each damage condition was measured five times. Therefore, any three sets of acceleration response data from a small number of sensors obtained from the simulation experiment are used as the training set for network training, and the data from the other two tests are used as the validation set and test set. At the same time, data augmentation needs to be performed after standardizing the acceleration response data. The lengths of the acceleration response data samples under different speeds can be calculated based on the sampling frequency, moving speed, and beam length, resulting in acceleration response datasets for the simulation model experiments under different speeds of 6000 × 8 and 12,000 × 8, respectively. From the above experimental conditions, the test beam model’s fundamental frequency is 10.4 Hz. Therefore, with a fundamental frequency of 10.4 Hz and a sampling frequency of 500 Hz, the length of the moving window *l* is calculated to be 97 (*l* = 1000/10.4). Since the input data for the variational autoencoder neural network model structure needs to be a power of 2 in length and to ensure the calculation data length, the final determined window length for data cropping is set to 256, with a moving step size of 32. Under this step size, the recognition effect of the variational autoencoder neural network model is almost the same as that with a larger step size.

Based on the calculated moving window length, data from a small number of sensors measuring the acceleration response signal is cropped, the experiment data only selects a few measuring points (four sensors), and the input dimension for the input layer is (256,4). The architecture of each neural network layer is shown in Table 6, with two additional convolutional layers added to the experimental data for deeper feature extraction. The dimension of the hidden layer is set to 500. The activation function of the variational autoencoder neural network still uses a combination of the Leaky ReLU function and Batch normalization function, with a learning rate set at 0.001.

The acceleration data from the test model is analyzed, and sensors are selected from positions 1, 3, 4, and 7 for data analysis. The acceleration data are shown in Figure 18. The experiment also analyzed the additional sensors extracted, and changes in sensor positions did not significantly affect the analysis results. The index chart analyzed for the undamaged condition is shown in Figure 19, indicating that there are no significant changes in the overall damage index of the structure under the undamaged state.

### 4.2. Experimental Results

The structure will be analyzed using the acquired acceleration data based on the experimental conditions and network settings mentioned above. This section will analyze different damage conditions.

#### 4.2.1. Single Damage Condition

Analyzing the single-damage condition, the well-trained VAE network model is utilized for identification. As is shown in Figure 20, when the cart speed is 0.25 m/s, regardless of whether it is a 10.5 kg or 20.5 kg cart, a significant drop and mutation in the damage factor of the VAE network occur near the damage position at 0.68 L, and it is below the threshold. Moreover, as the degree of damage deepens, the recognition index value will be lower, indicating poorer correlation and the network can accurately identify the location of the damage.

Under different speed conditions, at 0.5 m/s, the recognition indexes for each mass are shown in Figure 21. The VAE network can still accurately identify the location of the damage. The index values for both conditions are below the threshold, but index values are still interfering at the end. The excessively high speed causes abnormal values to be too large when passing through the bridge, but these do not exceed the index of the abnormal position where damage occurs, so the damage location can still be determined.

#### 4.2.2. Multiple Damage Conditions

The previous section analyzed the results of a single damage condition, and this part will analyze the experiments of multiple damage conditions.

Using the VAE network method to analyze the experimental model under the condition of double damage, it can be seen that the result graphs of MDC1 and MDC2 can display the damage locations well in Figure 22 and Figure 23. Regardless of whether it is at a speed of 0.25 m/s or 0.5 m/s, larger mutations are found at the positions of 0.38 L and 0.68 L, and the damage indicators are all below the threshold. The VAE network can accurately determine the locations of the two damages. The locations can still be accurately identified even when the conditions are under different masses. The larger the mass, the more obvious the feedback from the indicator graph, indicating that this method can still accurately judge the overall structural correlation and thus infer the state of the structure even when there are multiple damage anomalies.

## 5. Conclusions

This article addresses the challenge of structural health monitoring damage identification methods at present stage, where it is difficult to effectively identify the structural state without an accurate model and baseline data. A damage identification method is proposed based on the correlation of a small number of sensors using Variational Autoencoder neural networks. Some conclusions are summarized as follows:This paper proposes a damage detection and identification method based on the correlation analysis of limited sensor information using VAE neural networks. Utilizing the characteristics of moving loads and vehicle-bridge coupling, along with the principle of correlation, this method applies VAE neural network correlation analysis for damage detection and identification on simply supported beam models under moving loads;This method utilizes a small number of sensors and does not require non-destructive data to perform damage identification, effectively addressing the challenges in engineering applications where there are a large number of sensors and long service times without baseline data;VAE neural networks, grounded in deep learning networks, align well with the current challenge in structural health monitoring systems where massive data cannot be utilized timely and effectively, showing significant potential for practical development;The method proposed in this study, based on the correlation analysis of a small amount of sensor information using a variational autoencoder neural network, only utilizes a deep Bayesian network to construct the variational autoencoder model. However, there are still limitations in the model structure, such as insufficient training time for convenience and rapidity and the need to establish different network models for different structures to converge, which is not a one-size-fits-all solution. Additionally, it is necessary to conduct data testing on the application model or bridge to establish a certain model network for better recognition and analysis. Furthermore, when the number of sensors is further reduced, the network’s recognition results may become unstable. Therefore, there is room for further improvement and enhancement of this method by utilizing new deep learning methods for better application in practical engineering scenarios.

## Figures and Tables

**Figure 1 sensors-24-02616-f001:**
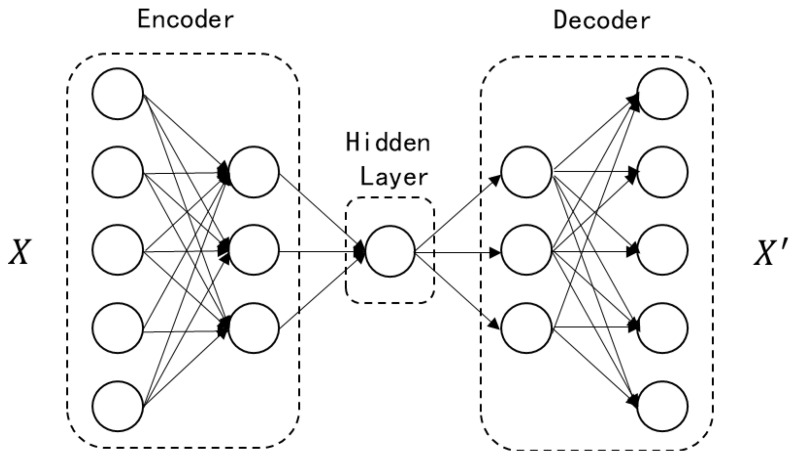
Autoencoder Neural Network Architecture Diagram.

**Figure 2 sensors-24-02616-f002:**
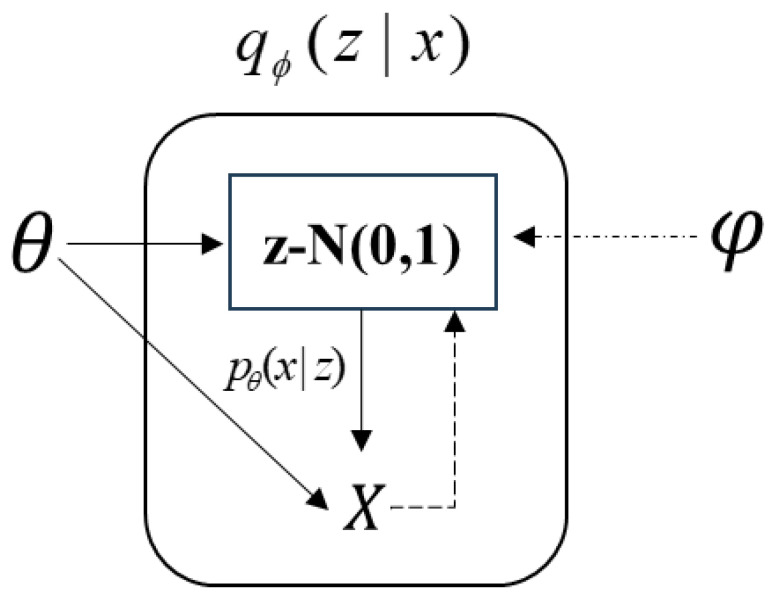
Internal operation of VAE.

**Figure 3 sensors-24-02616-f003:**
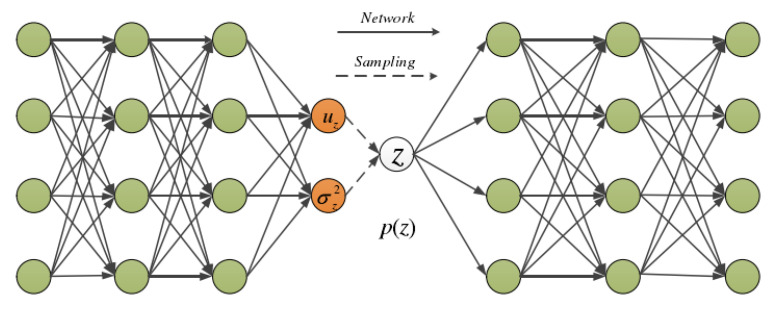
The structure of VAE.

**Figure 4 sensors-24-02616-f004:**
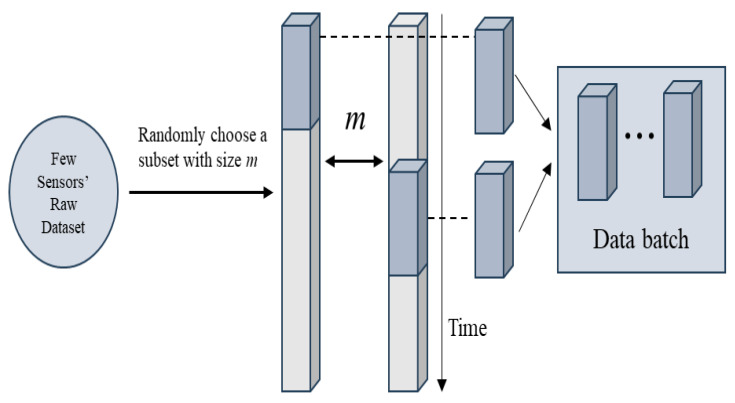
Data augmentation process.

**Figure 5 sensors-24-02616-f005:**
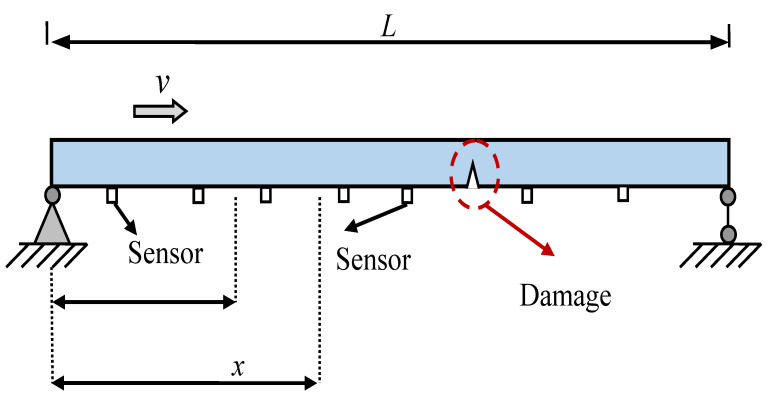
T Simply Supported Beam Model.

**Figure 6 sensors-24-02616-f006:**
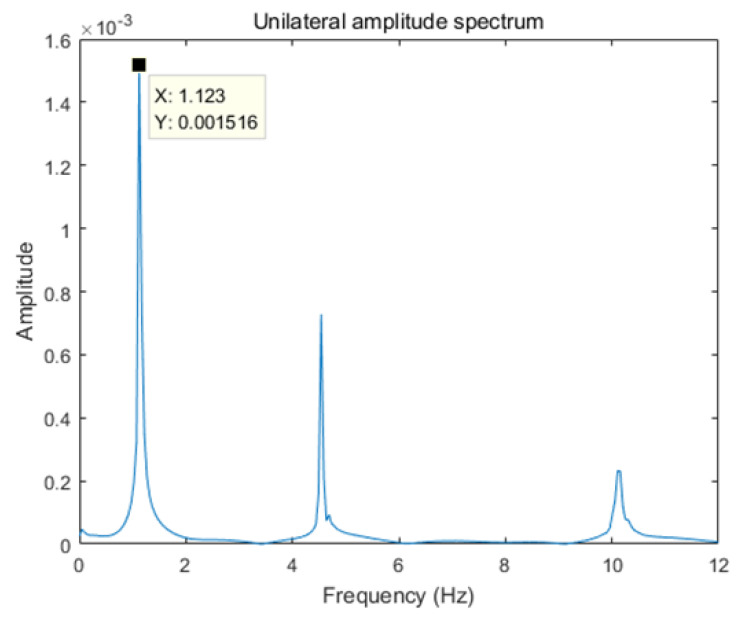
Fourier spectrum of the acceleration response.

**Figure 7 sensors-24-02616-f007:**
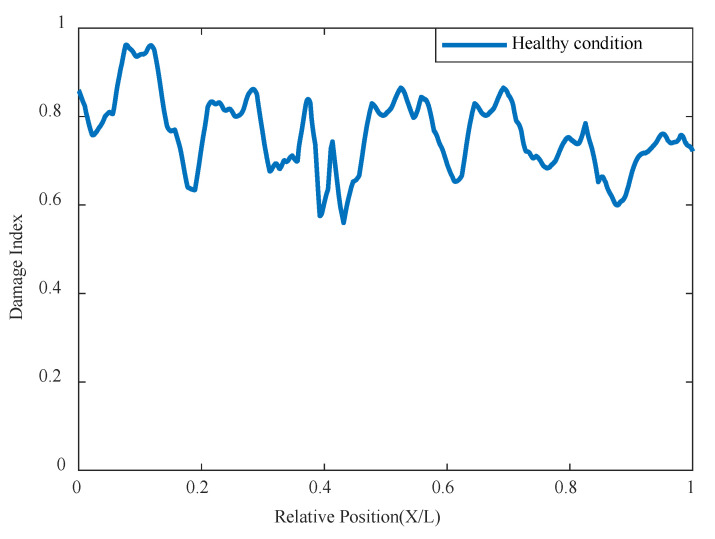
Damage factor curves in healthy status.

**Figure 8 sensors-24-02616-f008:**
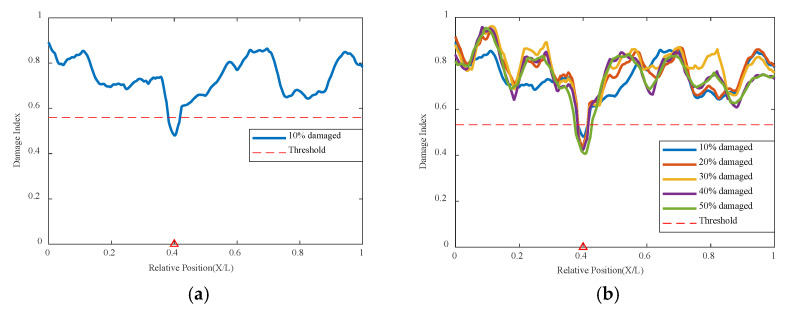
Damage factor curves under different damage severities with mass moving at 0.2 m/s: (**a**) γ = 10%, (**b**) γ = 10% to γ = 50% (The red triangle marks the location of the damage).

**Figure 9 sensors-24-02616-f009:**
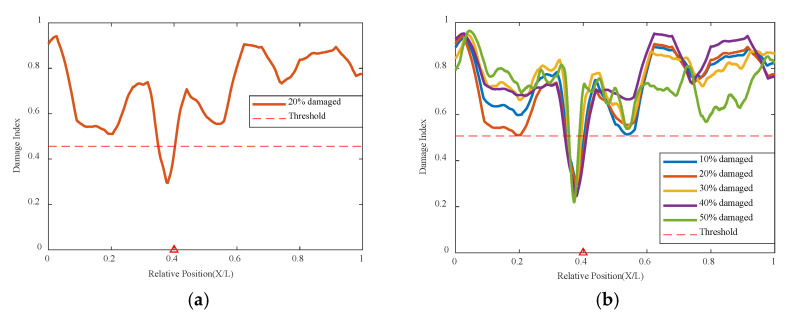
Damage factor curves under different damage severities with mass moving at 0.5 m/s: (**a**) γ = 20%, (**b**) γ = 10% to γ = 50% (The red triangle marks the location of the damage).

**Figure 10 sensors-24-02616-f010:**
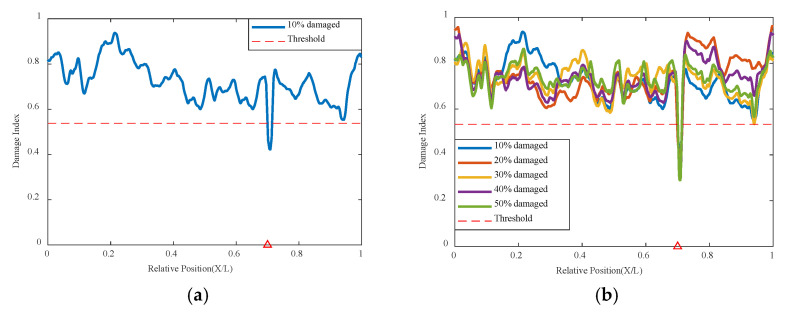
Damage factor curves under different damage severities with mass moving at 0.7 L damage location and 0.2 m/s: (**a**) γ = 10%, (**b**) γ = 10% to γ = 50% (The red triangle marks the location of the damage).

**Figure 11 sensors-24-02616-f011:**
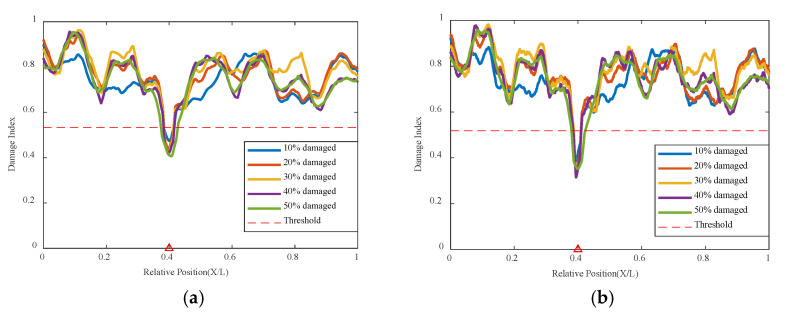
Damage factor curves for single damage detection under different noise: (**a**) γ = 10% to 50%, SNR =40 dB, (**b**) γ = 10% to 50%, SNR = 25 dB (The red triangle marks the location of the damage).

**Figure 12 sensors-24-02616-f012:**
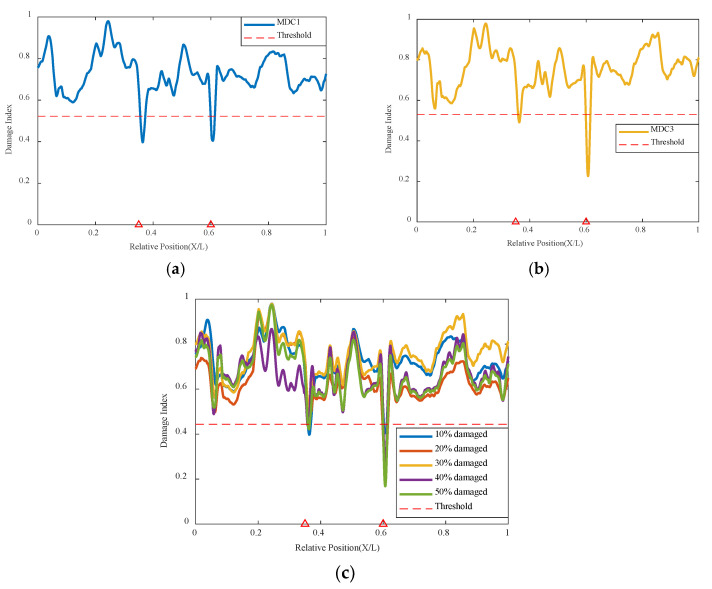
Damage factor curves for multiple damage detection under different damage severities: (**a**) MDC1, (**b**) MDC3, (**c**) MDC1 to MDC5 (The red triangle marks the location of the damage).

**Figure 13 sensors-24-02616-f013:**
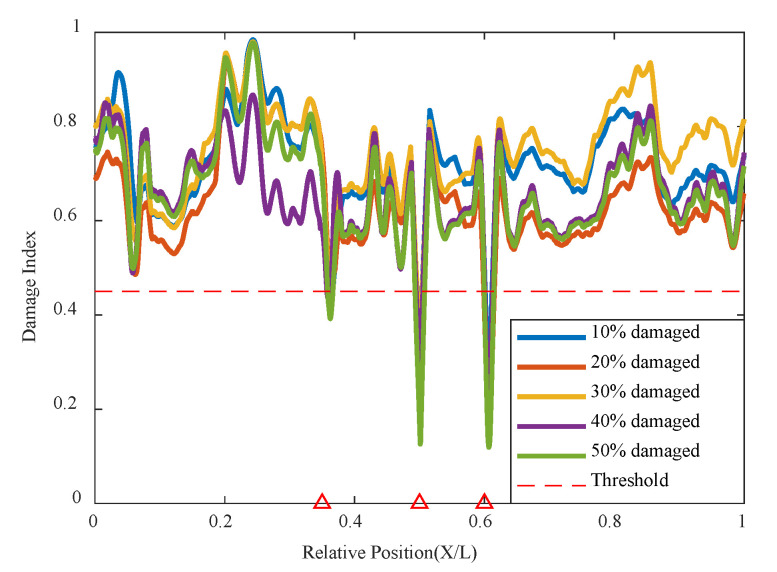
Damage factor curves for multiple damage detection under three damage conditions (The red triangle marks the location of the damage).

**Figure 14 sensors-24-02616-f014:**
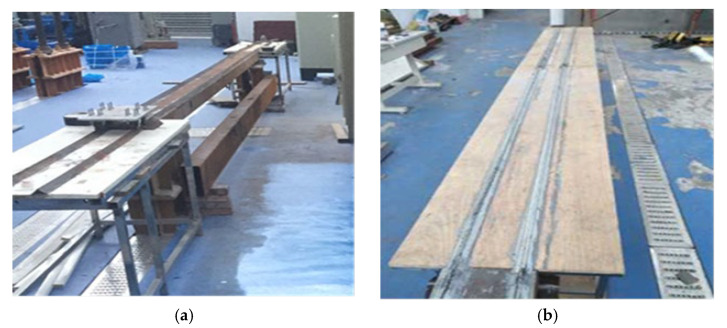
Experimental setup: (**a**) Experimental structure, (**b**) constant-speed bridges.

**Figure 15 sensors-24-02616-f015:**
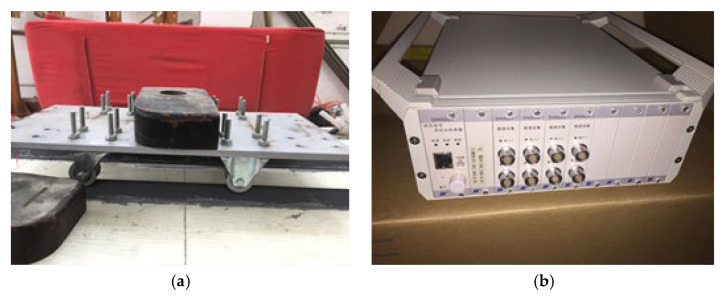
Experimental setup: (**a**) model car, (**b**) DH5920 dynamic test system.

**Figure 16 sensors-24-02616-f016:**
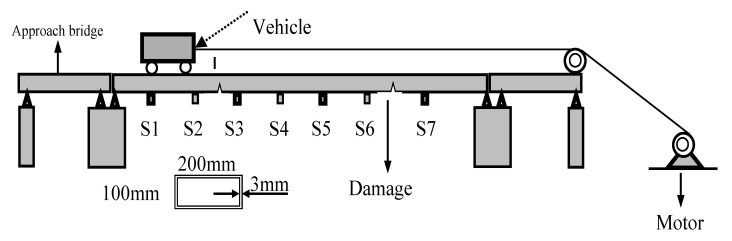
Experimental diagram.

**Figure 17 sensors-24-02616-f017:**
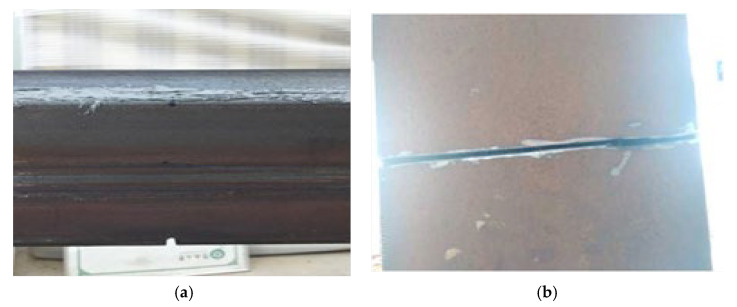
Vertical view of experimental damage: (**a**) side view of the damage fracture, (**b**) top view of the bottom damage.

**Figure 18 sensors-24-02616-f018:**
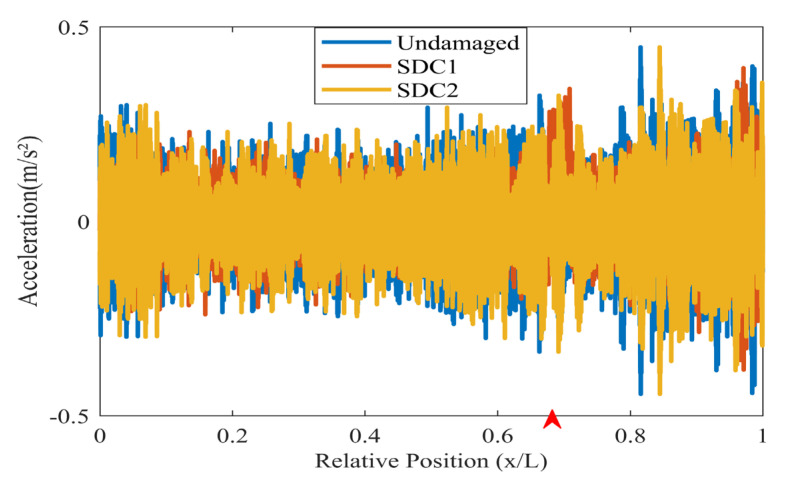
Acceleration data of experiment (The red arrow marks the location of the damage).

**Figure 19 sensors-24-02616-f019:**
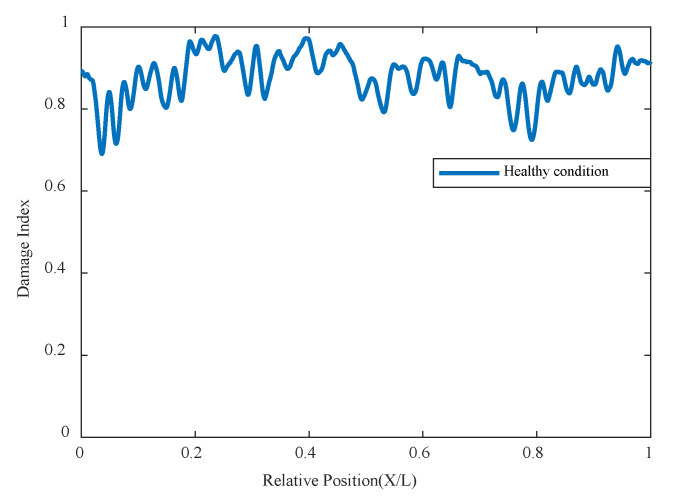
Damage factor curves calculated from the acceleration responses in healthy status.

**Figure 20 sensors-24-02616-f020:**
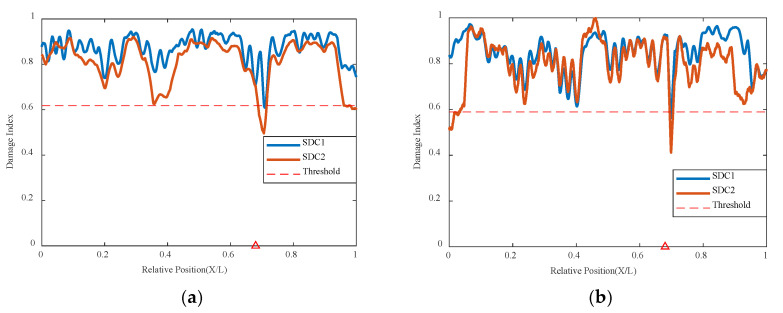
Damage factor curves for single damage detection under different damage severities: (**a**) mass with 10.5 kg and the speed with 0.25 m/s, (**b**) mass with 20.5 kg and speed with 0.25 m/s (The red triangle marks the location of the damage).

**Figure 21 sensors-24-02616-f021:**
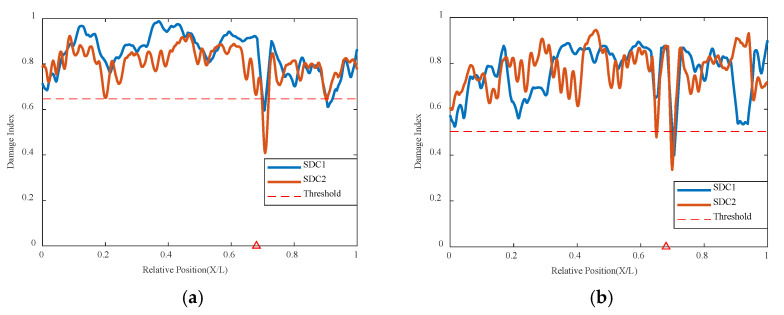
Damage factor curves for single damage detection under different damage severities: (**a**) mass with 10.5 kg and the speed with 0.5 m/s, (**b**) mass with 20.5 kg and speed with 0.5 m/s (The red triangle marks the location of the damage).

**Figure 22 sensors-24-02616-f022:**
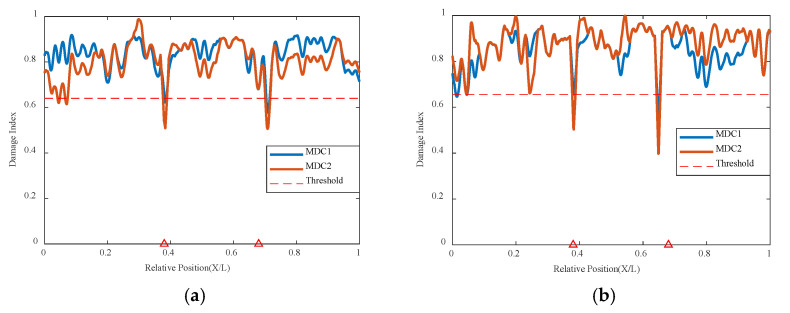
Damage factor curves for mutiple damage detection under different damage severities: (**a**) mass with 10.5 kg and the speed with 0.25 m/s, (**b**) mass with 20.5 kg and speed with 0.25 m/s (The red triangle marks the location of the damage).

**Figure 23 sensors-24-02616-f023:**
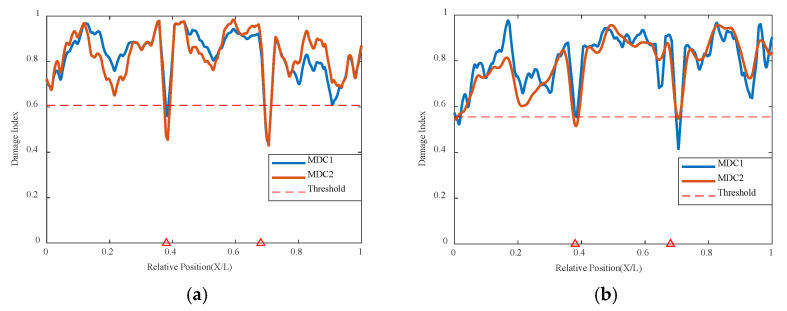
Damage factor curves for mutiple damage detection under different damage severities: (**a**) mass with 10.5 kg and the speed with 0.5 m/s, (**b**) mass with 20.5 kg and speed with 0.5 m/s (The red triangle marks the location of the damage).

**Table 1 sensors-24-02616-t001:** Ansys parameters.

Project	Parameters
Element type	Plane 42
Beam length	20 m
Elastic Modulus	200 GPa
Density	7.85 × 103 kg/m^3^
Section size	0.2 m × 0.1 m
Poisson ratio	0.3

**Table 2 sensors-24-02616-t002:** Single damage setting of numerical simulation.

Damage Location	Damage Scenarios	Speed (m/s)	Sampling (Hz)
0.4 L	γ = 10–50%	0.2,0.5	200
0.7 L	γ = 10–50%	0.2,0.5	200

**Table 3 sensors-24-02616-t003:** Mutiple damage setting of numerical simulation.

Damage Location	Damage Scenarios				
	MDC1	MDC2	MDC3	MDC4	MDC5
0.35 L	γ = 10%	γ = 10%	γ = 10%	γ = 10%	γ = 10%
0.6 L	γ = 10%	γ = 20%	γ = 30%	γ = 40%	γ = 50%

**Table 4 sensors-24-02616-t004:** VAE structure of the numerical simulation.

	Layer (Type)	Output Shape	Kernel Size/Pooling
Encoder	Input_layer	(1024,4)	/
	Conv1D_1	(1024,4)	4
	Maxpooling1D_1	(512,4)	4
	Conv1D_2	(512,8)	4
	Maxpooling1D_2	(256,8)	2
	Conv1D_3	(256,12)	4
	Maxpooling1D_3	(128,12)	2
	Flatten_1	1536	
Hidden layer	Dense_1	(428)	/
Decoder	Reshape	(128,12)	/
	Upsampling1D_1	(256,12)	2
	Conv1D_4	(256,12)	4
	Upsampling1D_2	(512,12)	2
	Conv1D_5	(512,8)	4
	Upsampling1D_3	(1024,8)	4
	Conv1D_6	(1024,8)	4
	ConvlD_7	(1024,4)	4

**Table 5 sensors-24-02616-t005:** Model experimental beam damage condition.

Damage Location	Damage Scenarios			
	SDC1	SDC2	MDC3	MDC4
*L*d = 0.38	-	-	4 mm	7 mm
*L*d = 0.68	3 mm	5 mm	7 mm	7 mm

**Table 6 sensors-24-02616-t006:** VAE structure of the experimental verifications.

	Layer (Type)	Output Shape	Kernel Size/Pooling
Encoder	Input_layer	(256,4)	/
	Conv1D_1	(256,8)	8
	Conv1D_2	(256,8)	8
	Maxpooling1D_1	(128,8)	4
	Conv1D_3	(128,12)	6
	Conv1D_4	(128,12)	6
	Maxpooling1D_2	(64,12)	4
	Conv1D_5	(64,16)	4
	Conv1D_6	(64,16)	4
	Flatten_1	(1024)	
Hidden layer	Hidden layer_1	(500)	/
	Hidden layer_2	(1024)	
Decoder	Reshape	(64,16)	/
	Conv1D_7	(64,16)	4
	Conv1D_8	(64,16)	4
	Upsampling1D_1	(128,16)	4
	Conv1D_9	(128,12)	6
	Conv1D_10	(128,12)	6
	Upsampling1D_2	(256,12)	4
	Conv1D_11	(256,8)	8
	Conv1D_12	(256,8)	8
	Conv1D_13	(256,4)	/

## Data Availability

Data are contained within the article.

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
