# Peer review of "Structural Damage Detection Based on the Correlation of Variational Autoencoder Neural Networks Using Limited Sensors"

_sensors, 2024, doi:10.3390/s24082616_

Round 1

Reviewer 1 Report

Comments and Suggestions for Authors

The paper proposed a method to detect structural damage using the Correlation of Variational Autoencoder Neural Networks with considering limited sensors. Variational Autoencoder Neural Networks are constructed to detect damage for acceleration data. Monitoring data obtained from an experimental simply supported beam bridge is used to validate the effectiveness of the approach. The topic is interesting. However, to improve academic quality and persuasiveness, this article should be major advised before being accepted. BTW, the writing style of the whole paper should be polished, as well as some obvious writing errors. Some specific comments are shown as follows:

1) It is noted that your manuscript needs careful editing by someone with expertise in technical English editing paying particular attention to English grammar, spelling, and sentence structure so that the goals and results of the study are clear to the reader.

2) The format should be uniform throughout the manuscript. Please verify and correct this. For example, in keywords “varia-tional autoencoder neural networks;”, No hyphen should be added here.

3) When proper noun abbreviation first appears, the full name should be given at the same time. For example, “MPCA”, “ICS”, “MDC”.

4) The introduction section of this paper lack of clarity and logic. Sentences are lacking focus and transitions. The author should organize the introduction content to make it more logical and persuasive. What’s more, relevant research background needs to be supplemented in introduction.

5) The figures in your paper are a bit blurry and uncomplete. Please consider replacing them with clearer ones.

6) In numerical simulations and experimental structures, the authors consider different damaged conditions. Why do different damaged conditions correspond to different thresholds, the threshold for healthy status should constant. Please interrupt the threshold in detail.

7) In section 3.2.2. Network Model Parameter Settings Table 4, the authors introduce the structure of VAE. It is constituted by Convolution layer, Maxpooling layer, Upsampling layer. But in the following paragraphs, the authors mention that deconvolution and unpooling layers are also used to restore the data. Please check carefully and correct it.

8) Generally speaking, the training of a network requires training set, valid set, and test set. Firstly, the neural network is trained through the training and valid set. After obtaining a good performance of the network model, the data is predicted and the results are analyzed according to the test set. In section 3.2. Finite Element VAE Network Configuration, the overall architecture of the VAE network is only divided into two parts: training and testing.

9) The effectiveness and robustness of the proposed approach are verified in this paper. This process is necessary and meaningful. In reviewer’s opinion, to validate the feasibility of the proposed approach in this paper, authors can further add the other condition of limited sensors and multiple damaged case. e.g., triple damages.

10) Clearly highlight the novelty and innovations of your manuscript in the conclusion section.

Comments on the Quality of English Language

The writing style of the whole paper should be polished, as well as some obvious writing errors.

Reviewer 2 Report

Comments and Suggestions for Authors

This paper presents a pioneering approach to structural damage identification utilizing Variational Autoencoder (VAE) neural networks based on correlation analysis of limited sensor data. The method offers a promising solution for identifying structural damage without the need for baseline data, thereby reducing sensor costs and improving practicality in engineering applications. Through the automatic extraction of features from signals using unsupervised learning, the VAE neural network model efficiently processes response data from a small number of sensors for accurate structural state determination. The efficacy of this method is validated through numerical simulations and physical experiments on beam structures, demonstrating its ability to pinpoint damage locations under various conditions., this study presents an approach for damage identification by integrating correlation analysis with VAE neural networks.

1. In part 2.3.1, "Since the data obtained itself has unit restrictions, all the data simulated in this paper are acceleration data. " "Data preprocessing allows dimensionless or data with different units to be better compared without affecting the intrinsic feature information of the data." According to the above description, since the simulation data used in this article are acceleration data, do these acceleration data have the same units? Are these acceleration data acceleration responses? If these data have the same units, according to the description of the second sentence, cannot be processed by normalization? Is there any reference for the normalization of acceleration data?

2. In lines 266-267, "The acceleration response data obtained from the simulation experiments are abundant, and each damage condition measurement is repeated five times." Why is each damage state measurement repeated 5 times? Can it be more or less? "The augmented data still retains the structural feature information, so 70% of the augmented data is used as the training set, and the remaining 30% is equally divided into the validation and testing sets." According to the description of the sentence, 70% of the data is used for the training set, and the measurement is repeated 5 times in each damage state. Why is the training set finally given 3 sets instead of 4 sets?

3. In lines 301-302, "According to the author's research findings, when a moving load passes through a damaged location on a bridge, it leads to a deterioration in correlation at that point." Are there any relevant or previous studies for reference here?

4. In lines 369-371, "Moreover, by controlling the calculation step size to reduce training time, a too small step size would lead to excessively long training and computation times, hence this paper sets the step size to 32." What happens if more than 32 steps are used in this study? For example, step size 60,100 or larger? Does it only affect the training time? How does it affect?

5. Several relative studies should be mentioned: Fiber optic health monitoring and temperature behavior of bridge in cold region, Structural Control and Health Monitoring. Damage identification of large-scale space truss structures based on stiffness separation method, Structures. Optimal static strain sensor placement for truss bridges, International Journal of Distributed Sensor Networks.

6. In lines 427-428, "The setting of the threshold is as described in the formula (10) in the previous section, …" formula (10) doesn't seem to be in the previous section. It is recommended to change the related description. Please describe the threshold setting theory and related formulas in detail.

Reviewer 3 Report

Comments and Suggestions for Authors

This paper presents an innovative approach to the challenge of structural health monitoring and damage identification methods, particularly in contexts where accurate models and baseline data are lacking. By proposing a damage identification method that utilizes Variational Autoencoder (VAE) neural networks to analyze the correlation of data from a limited number of sensors, the study offers a significant contribution to the field. However, while the methodology and its application to simply supported beam models under moving loads demonstrate potential, there are areas within the research that could benefit from further refinement. Taking into account the following comments could not only strengthen the arguments presented but also enhance the paper's applicability to practical scenarios.

  1. The structure of the paper would benefit greatly from the addition of a concluding paragraph at the end of Section 1. This paragraph should outline the paper's organization and provide a brief overview of what readers can expect in each subsequent section.
  2. To enhance the paper's structure and readability, each section should begin with an introductory sentence that outlines its content. Avoid immediately following a section introduction with a subsection (for example, section 2 and subsection 2.1); rather, start with a brief summary to guide the reader. This improvement will facilitate better navigation and understanding throughout the paper.
  3. I recommend that the authors include a reference immediately following the sentence on the first page, line 37, starting with "The methods for identifying structural damage." This reference, available at https://doi.org/10.12989/acc.2021.12.4.355, could be especially useful for readers seeking further information on this topic.
  4. On page 2, line 87, it is crucial to clarify the acronym VAE for readers who might not be familiar with the term. It has been introduced at line 99. Please introduce it at its first usage.
  5. It is incorrect to define an acronym as "artificial neural network (ANN)"; instead, it should be "Artificial Neural Network (ANN)." Ensure consistency in the formatting of your acronym definitions throughout the paper.
  6. Improve the quality of Figures 3 and 6.
  7. Include a sentence analyzing Figure 18. The analysis is provided in a different section, but for better readability, it is preferable to have it near the evidence.
  8. You should omit Section 6, as filing a patent will not be feasible with the publication of the current paper.

Round 2

Reviewer 1 Report

Comments and Suggestions for Authors

After revision, the format, grammar, logic of this manuscript has been improved and related problems have been effectively corrected. It makes the method proposed in this manuscript more persuasive. However, there are some deficiencies in this paper. In the reviewer's opinion, this paper can be accepted for publication after making those revisions. The specific comments are shown as follows:

 1) There is at least one Spelling error in the manuscript, such as, in line 3, “Varia-tional” would be “Variational”, in line 554, “hree” would be “three”. Please check the manuscript carefully.

 2) The figures in your paper are a bit blurry. Please consider replacing them with clearer ones.

 3) In formula (6)-(9), parameters in the formulas lack annotations, and should be explained. Please check and correct it.

 4) Another obvious problem is that the explanation of the numerical simulation results is insufficient. That should be improved.

 5) In section 3 “VAE-Related Correlation Analysis Numerical Simulation”, the effect of noise is only considered during single damage condition. If the noise is also considered in multiple damages condition, does the network still show a high robustness?

 6) In this manuscript, the author utilizes response data from limited sensors for model training analysis to determine the structural state. But only one limited sensor case is considered. In the reviewer’s opinion, to validate the feasibility of the proposed approach, the authors need consider the influence of different number of sensors on damage detection.

Reviewer 2 Report

Comments and Suggestions for Authors

The manuscript has been revised and the quality has been improved.

Reviewer 3 Report

Comments and Suggestions for Authors

My second comment must have been used to fix the rest of the sections of the paper and not only one section. :Avoid immediately following a section introduction with a subsection ; rather, start with a brief summary to guide the reader. The issue is still present at subsection 2.3, subsection 3.1, subsection 4.1.

Subsection 3.2.1 starts with talking about a figure, where is the first sentence introducing the content of the subsection, pleae check all sections and all subsections. All of them should have a sentence introducing brifely their content. 

The authors should also include the limitation of their application in the conclusion section as well. 
